# Preparation and Study of a Simple Three-Matrix Solid Electrolyte Membrane in Air

**DOI:** 10.3390/nano12173069

**Published:** 2022-09-03

**Authors:** Xinghua Liang, Xingtao Jiang, Linxiao Lan, Shuaibo Zeng, Meihong Huang, Dongxue Huang

**Affiliations:** 1Guangxi Key Laboratory of Automobile Components and Vehicle Technology, Guangxi University of Science and Technology, Liuzhou 545006, China; 2China School of Automotive and Transportation Engineering, Guangdong Polytechnic Normal University, Guangzhou 510632, China; 3Guangdong Polytechnic of Industry and Commerce, Guangzhou 510550, China

**Keywords:** composite solid electrolyte, lithium ion conductivity, capacity retention, flame retardancy

## Abstract

Solid-state lithium batteries have attracted much attention due to their special properties of high safety and high energy density. Among them, the polymer electrolyte membrane with high ionic conductivity and a wide electrochemical window is a key part to achieve stable cycling of solid-state batteries. However, the low ionic conductivity and the high interfacial resistance limit its practical application. This work deals with the preparation of a composite solid electrolyte with high mechanical flexibility and non-flammability. Firstly, the crystallinity of the polymer is reduced, and the fluidity of Li^+^ between the polymer segments is improved by tertiary polymer polymerization. Then, lithium salt is added to form a solpolymer solution to provide Li^+^ and anion and then an inorganic solid electrolyte is added. As a result, the composite solid electrolyte has a Li^+^ conductivity (3.18 × 10^−4^ mS cm^−1^). The (LiNi_0.5_Mn_1.5_O_4_)LNMO/SPLL (PES-PVC-PVDF-LiBF_4_-LAZTP)/Li battery has a capacity retention rate of 98.4% after 100 cycles, which is much higher than that without inorganic oxides. This research provides an important reference for developing all-solid-state batteries in the greenhouse.

## 1. Introduction

All-solid-state batteries have received extensive attention due to the advantages of high safety and high energy density in recent years. The use of electrolyte membranes replaces traditional electrolytes and separators, acting as Li^+^ transport bridges in batteries [1,2]. The quality of the electrolyte membrane determines the cycle performance of the battery. Solid electrolytes for lithium battery development can generally be divided into two categories: ceramic-based solid-state electrolytes and polymer-based solid-state electrolytes [3,4]. In ceramic solid electrolytes, cation groups and metal cations form a unit cell skeleton, which provides a channel for the transmission of lithium ions. Each unit cell is connected to each other to form a network structure, which enables lithium ions to perform vacancy migration and interstitial migration inside to complete the transport of lithium ions [5]. Inorganic polymer electrolytes have thin films that can be prepared into thinner shapes—which can reduce interfacial impedance with positive and negative electrodes and can have good mechanical ductility—that can adapt to changes in battery conditions during charging and discharging. These materials have been studied for many years, but each has its own advantages and disadvantages [6,7]. The integration of material advantages is an important aspect of innovation. Xiang et al. prepared a Li/LiPON/LiCoO_2_ all-solid-state battery with a capacity loss of less than 2% after 4000 cycles. Subsequently, organic polymer solid electrolytes, inorganic solid electrolytes, and organic/inorganic composite solid electrolytes appeared in succession [8]. Polyethylene oxide (PEO) based polymer solid electrolyte has attracted much attention [9]. In the PEO-based polymer solid electrolyte, the conductive Li^+^ in the amorphous region coordinates with the ether oxide (EO) on the PEO segment and realizes Li^+^ migration through the complexation and decomplexation process of the lithium-oxygen bond (Li-O) movement on the PEO chain segment [10,11,12]. However, PEO has higher crystallinity and less lithium ion migration at room temperature, resulting in extremely low ion conductivity (only 10^−8^–10^−7^ S cm^−1^) [13]. The inorganic solid electrolyte has distinctive merits, including high chemical stability, environmentally friendly, high safety, high conductivity, wide electrochemical window, and good thermal stability. Li et al. [14], prepared a high-performance three-dimensional cross-linked electrolyte based on polyvinylidene fluoride (PVDF) and polyethylene oxide (PEO), which has good performance at room temperature, but the cycle performance needs to be improved. However, shortcomings, such as preparation and storage requirements and flexibility, restrict their development [15,16,17].

The organic/inorganic composite solid electrolyte was developed to have the conductivity of inorganic solid electrolytes and the flexibility of PEO-based polymer electrolytes, opening up new development prospects for all-solid-state lithium-ion batteries with high safety and a long cycle life [18,19]. However, its electrochemical performance is still worse than traditional liquid batteries [20,21,22]. Therefore, preparing a greenhouse with high ion channels and solving interface problems has high prospects.

Here, we propose the strategy of combining organic electrolytes and inorganic electrolytes. Firstly, the crystallinity of the polymer is reduced, and the fluidity of Li^+^ between the polymer segments is improved by tertiary polymer polymerization. Then, lithium salt is added to form a sol polymer solution to provide Li^+^ and anion and then an inorganic solid electrolyte is added. By adding inorganic fillers, the regular arrangement of polymer segments can be destroyed and made in an amorphous state to increase the amorphous region conducive to lithium ion transport. In addition, inorganic solid electrolytes have high greenhouse conductivity and a wide electrochemical window. Inorganic fillers can also reduce the activation energy of lithium ion migration to form channels conducive to lithium ion migration.

## 2. Materials and Methods

### 2.1. Preparation of LATP Powder

The solid phase method uses lithium carbonate (Li_2_CO_3_), aluminum oxide (Al_2_O_3_), titanium dioxide (TiO_2_), and ammonium dihydrogen phosphate (NH_3_H_2_PO_4_) zinc oxide (ZnO) as precursors. Each material is weighed according to Li_1.3_Al_0.3_Ti_1.7_(PO_4_)_3_(LATP) and Li1_.3_Al_0.1_Zn_0.1_Ti_1.8_(PO_4_)_3_(LAZTP). Calculate the molar ratio to prepare 40 g of LATP and LAZTP materials, respectively, for use in subsequent experimental tests. After mixing uniformly, ball mill for 8 h, use ethanol as the dispersant, and the speed of the ball mill is 280 rpm. After ball milling, the powder was dried in a drying oven at 80 ℃ for 10 h. After the powder was dry, it was processed by grinding. The powder was annealed and calcined in an atmosphere furnace at 950 ℃ for 4 h to obtain two white LATP and LAZTP precursor powders.

### 2.2. Preparation of Electrolyte Sheet

After the powder is ground through the grinding table, the two powders are pressed under a pressure of 18 MP for 5 min and pressed into a disc with a diameter of 16 mm and a thickness of 1 mm. The layer powder is placed in an atmosphere furnace, calcined at 900 °C for 4 h at a heating rate of 5 °C/min, and naturally cooled to room temperature to obtain high-temperature sintered LATP and LAZTP solid electrolyte sheets, which are then polished to the required thickness by sandpaper. All LATP and LAZTP electrolyte sheet sintered samples were ultrasonically cleaned in absolute ethanol for 10 min and then dried for 2 h. Conductive silver paste was applied to both sides and steel sheets were pasted and placed in a drying oven for 3 h for testing.

### 2.3. Preparation of SPLL Composite Electrolytes

Preparation of three-matrix solid electrolyte SPLL (PES-PVC-PVDF-LiBF_4_-LAZTP). First, add 30 mL of DMF solution in a 100 m beaker, stir it electrically at a temperature of 60 °C, then add 2 g of PES until the solution becomes a clear sol-like solution and then add 1g of PVA. When the mixture is stirred until there are no air bubbles, add 5% PVDF to form a copolymerized base, sonicate for 20 min to mix the three completely, then add LiBF_4_ (add with PES es: Li=8:1), and stir for 4 h. Then, add 10% LAZTP, ultrasonic for 30 min, stir for 10 h, pour it into a polytetrafluoroethylene container, dry in a blast drying oven at 60 °C for 12 h, then remove and cut it to a size of 16 mm, and put it in a glove box for later use. 

### 2.4. Preparation of Solid-State LFP/SPLL/Li Cell

The CR2032 button cell is assembled in an argon box using the above composite SSE in an argon-filled glove box. The LNMO cathode was prepared by mixing 80 wt.% of commercial LNMO (Shanghai Ales Co., Ltd.), 10 wt.% conductive carbon black, and 10 wt.% PVDF dissolved in NMP solvent. After thorough stirring, the cathode slurry was evenly cast onto the aluminum foil. Subsequently, the aluminum foil was dried under vacuum at 110 ℃ and then cut into circular electrodes with a diameter of 16 mm. In addition, lithium metal is used as the anode of assembled button batteries. The electrochemical performance of all assembled batteries is measured in the voltage range of 3.5–5 V.

### 2.5. Physical Characterizations

X-ray diffraction (XRD, D8-Advabce, Bruker, Frankfurt, Germany, in the range of 10–90 °) and Raman spectroscopy were used to analyze the phase structure of the positive electrode material. The surface structure of the material is observed through a field emission scanning electron microscope (SEM, Sigma04-55, ZEISS, HORIBA, Longjumeau, France). The electrochemical window test is carried out by linear sweep voltammetry (LSV) at a scanning rate of 0.1 mV·s^−1^. A stainless steel sheet (SS) is used as the working electrode, and a lithium sheet is used as the counter electrode. The LAGP glass and GC structural studies were carried out using 13 C Solid State Nuclear Magnetic Resonance (NMR) using BRUKER 700 MHz HD spectrometer on a 2.5 mm Trigamma probe at a spinning frequency of 32 kHz. The SPLL and SPL anode surface was characterized by X-ray photoelectron spectroscopy (XPS, Thermo Escalab 210 system, Dreieich, Germany).

### 2.6. Electrochemical Measurements

The ionic conductivity of the electrolyte can be obtained by the following equation:σ=L(R×S)
where *σ* represents the ionic conductivity, *L* is the thickness of the sample, *S* represents the contact area between the electrolyte and the test electrode (SS), and *R* is the resistance measured by impedance spectroscopy.

The lithium ion transference number ($t_{Li^{+}}$) is an important parameter for evaluating polymer electrolyte membranes. A higher number shows the lithium ion transfer strength of the membrane, and the transfer is more important during the cycle. Usually, the test is performed by the timing method on a lithium symmetrical battery (such as Li/SPLL/Li) at a voltage of 1 mV for 4000 s, and the value is calculated by the following equation.
$$t_{Li^{+}}=\frac{I_{s}\left(\Delta V-I_{0}R_{0}\right)}{I_{0}\left(\Delta V-I_{s}R_{s}\right)}$$
where *I*_0_ and Is are the current values at the beginning and after the DC polarization is stabilized, and *R*_0_ and *R_s_* are the impedance values before and after DC polarization, respectively, and ∆*V* is the voltage value acting on both ends of the battery [23]. 

The electrochemical stability window of the electrolyte membrane is obtained by linear scanning voltammetry. The lithium sheet is used as the counter electrode and the reference electrode, the stainless steel sheet is used as the working electrode (SS), and the electrolyte membrane is in the middle (Li/SPLL/SS). The test range is 0–6 V, and the sweep speed is 0.5 mV s^−1^.

The cycle rate test is a standard for evaluating the quality of a battery. The assembled battery is tested for the cycle and rate performance between the 3.5–5 V electrochemical window, using the Xinwei tester and the DH7000 (Shenzhen, China) tester for impedance and CV testing.

## 3. Results

It can be seen from Figure 1b that the diffraction peaks of the two samples are consistent with the standard card (35-0754) of the NASICON (Na^+^ superionic conductor) structure (R-3c). The three prominent diffraction peaks are sharp, indicating that the synthesized samples are of high crystallinity. Negligible spurious peaks indicate some purity of the LATP and LAZTP samples. The intensity of the diffraction peak at 1 is higher than at 2 in Figure 1b, meaning the successful incorporation of Zn element into the lattice system, and it causes vacancies or defects in the lattice in the LAZTP sample. The SEM-EDS images in Appendix A further confirmed that the Zn element is successfully doped with a small amount in the LAZTP sample.

As shown in Figure 1c, the conductivity of LAZTP is much higher than LATP (1.69 × 10^−3^ S cm^−1^ vs. 4.02 × 10^−4^ S cm^−1^). Theoretical calculations are used to confirm the further conductivity of LAZTP. The calculation results show that the energy bandgap of LATP is 0.165 eV (Figure 1d), while the energy bandgap of LAZTP (Figure 1g) is only 0.05 eV. Such a small bandgap is beneficial for electrons to transition from the valence band into the conduction band and improve the electronic conductivity of LAZTP.

The density of states of the two samples is shown in Figure 1e,h. LAZTP is metalized after doping. The energy transition decreases and the conductivity increases, which is consistent with the experimental results, indicating that the doping of Zn element greatly improves the conductivity of the material.

As shown in Figure 1f,i, the particle distribution of the samples is relatively uniform with low agglomeration and clear boundary. Compared with LATP, the LAZTP has a smaller grain size and a smoother surface, which is conducive to the migration of Li^+^.

The interaction mechanism between lithium ions and electrolyte skeleton was further revealed by XRD and Raman spectroscopy. As shown in Figure 2a, the XRD patterns of (Polyethersulfone)PES, (Polyvinyl chloride)PVC, and PVDF are different, and their characteristic peaks are displayed in the SP (PVDF-PES-PVC) polymer. The diffraction peak at ≈20° is broad, indicating that amorphous regions are formed in the SP polymer, conducive to the transmission of Li^+^. No impurity peaks can be observed in the XRD patterns, suggesting the high purity of the composite electrolyte. Figure 2b shows the Raman spectra of the polymers in the range of 2000–3000 cm^−1^. The characteristic peaks of the SP polymer are only the superposition of the peaks of the three polymers, indicating that the SP polymer is only copolymerized of the three polymers, which is in good agreement with XRD results. Figure 2c shows the XRD patterns of SP, SPL, and SPLL. Compared with SP, the peaks at 2θ ≈ 20° and ≈33°of SPLL are significantly weakened, indicating that with the addition of LiBF_4_, the heterogeneous salt doping process changes the locally ordered polymer-ion assembly, filling the defective LiBF_4_ nanocrystalline grains into the intercrystalline network, changing the lattice structure, and improving the conductivity of the copolymer substrate. The peak intensities of SPLL are different from SPL, which is due to the addition of LAZTP in SPLL. Figure 2d shows the Raman spectra of SP, SPL(PVDF-PES-PVC-LiBF_4_), and SPLL, which reflects that the structure of SP changes with the addition of LiBF_4_, and the structure of SPL changes with the addition of LAZTP.

Electrochemical impedance spectroscopy (EIS) was utilized to explore the conductivity of the composite electrolytes [24]. As shown in Figure 2e, the conductivity of SPL polymer electrolytes increases as the temperature rises. However, for SPLL polymer electrolyte, the conductivity did not dramatically change, indicating that the electrolyte membrane is adapted to a broad temperature environment and is more stable than SPL. Figure 2f shows that the interface impedance decreases as the temperature changes for SPLL. Still, the phase does not change, indicating that the prepared electrolyte membrane can maintain stable cycling at different temperatures.

Figure 3 shows the preparation flow chart of the polymer electrolyte. PES, PVC, and PVDF polymers constitute a polyelectrolyte and then lithium salt LiBF_4_ is added to form a stable polymer. LAZTP is used as an additive to improve lithium ion transport channels, and inorganic metal oxides are added to increase the amorphous area of the polymer. After mixing uniformly, the composite is poured on a polytetrafluoroethylene template. Subsequently, the composite is cut to the size of a steel sheet after drying. Afterward, the product is assembled into a quasi-solid battery. Then, electrolyte was added to the battery to improve the humidity of the composite interface. The battery was taken out from the glove box and put in a dry box to stabilize polymerization. Finally, the battery was tested under air conditions.

Figure 4a shows the assembly sequence of the prepared electrolytes in the solid-state battery. The lithium ion migration number of SPL is calculated by chronoamperometry combined with AC impedance spectroscopy. At the initial stage of polarization, both Li^+^ and LiBF_4_ provide current. At the end of polarization, only Li^+^ are transferred from one lithium electrode to another, and the current reaches a constant value. In SPL polymer electrolytes, the mobility of Li^+^ is lower than that of the corresponding anions due to the high complexation of Li^+^ with the three polymer substrates. As shown in Figure 4b,e, the Li+ transference number of SPL is 0.32, while the lithium ion migration number of SPLL is 0.5, which is significantly higher than that of SPL. It was believed that the high lithium ions mobility of SPLL is due to the binding effect of LAZTP on anions.

The electrochemical stability window of the composite electrolytes was carried out by linear sweep voltammetry. As shown in Figure 4d, when the voltage rises to 5.1 V, the SPL is oxidized and decomposed, while the electrochemical stability window of SPLL can increase to more than 5.2 V, indicating that adding LAZTP to the polymer can increase the electrochemical stability window of polymer electrolyte. The wide electrochemical stability window of the composite electrolyte system is due to the strong interaction between small molecules and the trapping effect of a large number of micropores [25]. Therefore, SPLL can match the high voltage cathode of high-energy-density lithium-ion batteries.

The structural stability of the composite electrolytes was evaluated in the air. The indoor relative humidity is 40%. XRD was used to analyze the change of crystal structure. As shown in Figure 4g, new peaks at 2θ ≈ 36° and ≈55° were produced after 40 min, indicating that slight water decomposition occurred in the SP polymer. As shown in Figure 4h, new peaks appeared after 40 min, corresponding to Li_2_O and LiOH, indicating that the crystal structure of SPL polymer electrolyte changed. As shown in Figure 4i, the peak strength of SPLL polymer does not change significantly, indicating that the addition of LAZTP improves the stability of the composite electrolyte.

Figure 5 presents the chemical composition of the polymer electrolyte by X-ray photoelectron spectroscopy (XPS). Figure 5a represents the C element of SPL and SPLL. The peaks of SPLL at the low value are higher than SPL, indicating that the internal C structure of the polymer has changed after adding LAZTP and SiO_2_, resulting in highly stable under air circulation. As shown in Figure 5b, for SPLL, two peaks at 402.3 eV and 399.8 eV in the pristine N1s spectrum correspond to the N in [bmim]^+^ cation (Ncation) and [Tf_2_N]^−^ anion (Nanion). Compared with SPL, the N element peaks of SPLL increased significantly, indicating that the number of anion groups increased, which accelerated the transport of lithium ions. As shown in Figure 5c, the intensity of the Li^+^ peak is weak, suggesting an increase of the amorphous region area in the polymer, thereby increasing the Li^+^ mobility of the polymer.

To prove that the addition of LAZTP and SiO_2_ reduces the crystallinity of polymer electrolyte and provides a lithium ion migration channel, we analyzed the local chemical environment of lithium ion by ^13^C solid-state nuclear magnetic resonance (NMR). As shown in Figure 5d, the peaks of SPL are scattered, and there are many impurity peaks. In contrast, for SPLL, the peaks near 50 ppm and 140 ppm gradually become sharp, indicating that the addition of LAZTP and SiO_2_ reduces the crystallinity of SPLL polymer and enhances the conductivity of the material, which is consistent with the EIS results [26].

Constant current charge and discharge (GCD) were used to evaluate the influence of the addition of LAZTP and SiO_2_ on the electrochemical stability of the polymer electrolytes under the current density of 0.05, 0.1 and 0.2 mA cm^−2^. Figure 6c shows that the Li/SPLL/Li battery exhibits a stable Li electroplating/stripping under the corresponding current density. Moreover, it displays excellent stability even after 1000 cycles of long-term plating/stripping, demonstrating a good rate performance. On the contrary, the SPL polymer appears flocculent at 0.2 mA cm^−2^ during the plating/stripping process (Appendix A) [27]. Figure 6b shows that the interface impedance increases after the cycling test, indicating that a stable SEI film is formed at the interface, reducing the loss at the interface and improving the Li^+^ transmission rate. According to literature reports, adding inorganic metal oxide LAZTP [28] and non-metal oxide SiO_2_ [29] can increase the conductivity and amorphous region of the polymer electrolyte and improve the lithium ion migration.

To evaluate the electrochemical stability of the electrolytes, the LNMO/SPLL/Li and LNMO/SPL/Li batteries were assembled with SPLL and SPL electrolyte membranes, respectively [30]. Figure 7a,d shows the rate performance of the two batteries at 0.1 C, 0.2 C, 0.3 C, 0.5 C, 1.0 C, and 2.0 C. Compared with the SPL polymer battery, the SPLL polymer electrolyte battery exhibits a better rate performance. Appendix A and Figure 7b are the cycle performance and CV curves of the LNMO/SPLL/Li battery. The CV curves change slightly after five cycles, indicating that the battery has a stable cycle performance [31]. Figure 7c displays the discharge/charge curves of the first 100 cycles at a rate of 0.1 C at room temperature. It has a high voltage discharge platform of 4.7 V and a high cycle stability. The charge and discharge specific capacity of the first cycle are 140 mAh/g and 122 mAh/g, respectively, which are close to the theoretical specific capacity of LNMO, suggesting an excellent cycle stability of the SPLL polymer [32]. Figure 7e shows the Nyquist plots of the LNMO/SPLL/Li battery after various charging/discharging cycles at the fully charged state. The charge transfer impedance decreases and finally stabilizes with the increase of the cycle number, indicating that the SPLL polymer has a good cycle performance. As shown in Figure 7f, the capacity retention rate of the LNMO/SPLL/Li battery is still 98.4% after 100 cycles. On the contrary, the LNMO/SPL/Li battery displays a poor cycle performance (Appendix A), indicating that adding inorganic metal oxide LAZTP and non-metallic oxide SiO_2_ could improve the stability of the polymer electrolyte

The morphology of the SPLL polymer electrolyte was characterized by SEM before and after cycling. The electrolyte exhibits a smooth surface morphology before cycling (Appendix A) and shows a cross-networking morphology after 100 cycles, indicating that the polymer electrolyte membrane reacts to form an ion conduction channel for lithium ion transmission. It can also be seen from the cross-section that the uncirculated electrolyte cross-section is neatly arranged and dense. When electrochemical charging and discharging are carried out, a network structure appears inside the electrolyte, which indicates that a chemical reaction occurs internally during the charging and discharging process, forming a new type of electrolyte. The conductive network provides a powerful channel for lithium ion transmission.

Appendix A shows the morphology of the SPLL polymer electrolyte membrane before and after cycling. The electrolyte displays a white morphology before cycling. However, the surface of the electrolyte turns black after 100 cycles, which is the residue of the LNMO cathode. At the polymerization temperature (60 °C), SPLL is in close contact with LNMO. The good interface contact ability is conducive to the shuttle of Li+ in the interface. Appendix A shows the XRD patterns of the polymer electrolyte membrane before and after cycling. The characteristic peak is very sharp before cycling, and no by-products were generated after 20 cycles, indicating that no side reactions occurred during cycling.

The cycling stability of the LNMO/SPLL/Li batteries is evaluated at a current density of 0.25 C. As shown in Figure 8, the battery can reach up to 117.5 mAhg^−1^ in the first cycle. After 500 cycles, the capacity can still reach 85.6 mAhg^−1^ with a capacity retention rate of 82.5% [33]. For such a good performance, we attribute it to several aspects. First, the LAZTP can provide lithium ion transport channels to improve the ionic conductivity of the polymer. On the other hand, the non-metal oxide SiO_2_ can increase the amorphous area of the polymer and expand the lithium ion transport channel. Finally, we add a small amount of electrolyte on both sides of the polymer SPLL to reduce interface impedance.

The polymer electrolyte is ignited (Appendix A) to test its stability in the air. The electrolyte membrane burns when the fire source touches the electrolyte for 2 s. Subsequently, the electrolyte membrane goes out when being removed from the fire source. When we again put it on the polymer electrolyte for 4 s, the electrolyte membrane burned up, and the electrolyte membrane extinguished immediately as the fire source was removed. These results show that the electrolyte has a good air stability.

## 4. Conclusions

This work prepared a composite solid electrolyte with high mechanical flexibility and non-flammability. Firstly, the crystallinity of the polymer is reduced, and the fluidity of Li^+^ between the polymer segments is improved by tertiary polymer polymerization. The composite solid electrolyte has an excellent Li^+^ conductivity (3.18 × 10^−4^ mS cm^−1^). The LNMO/SPLL/Li battery has a capacity retention rate of 98.4% after 100 cycles, which is much higher than that without inorganic oxides. This research provides an important reference for developing all-solid-state batteries in a greenhouse.

## Figures and Tables

**Figure 1 nanomaterials-12-03069-f001:**
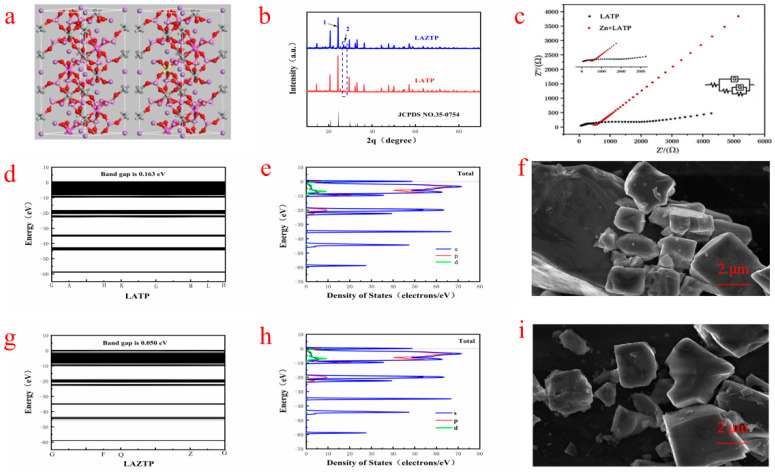
(**a**) LATP and LAZTP unit cell model. (**b**) XRD patterns of LATP and LAZTP. (**c**) Nyquist patterns of LATP and LAZTP. (**d**,**e**) Band structure and total density of states diagrams of LATP. (**g**,**h**) Band structure and total density and states diagrams of LAZTP. (**f**) SEM image of LAZTP. (**i**) SEM image of LATP.

**Figure 2 nanomaterials-12-03069-f002:**
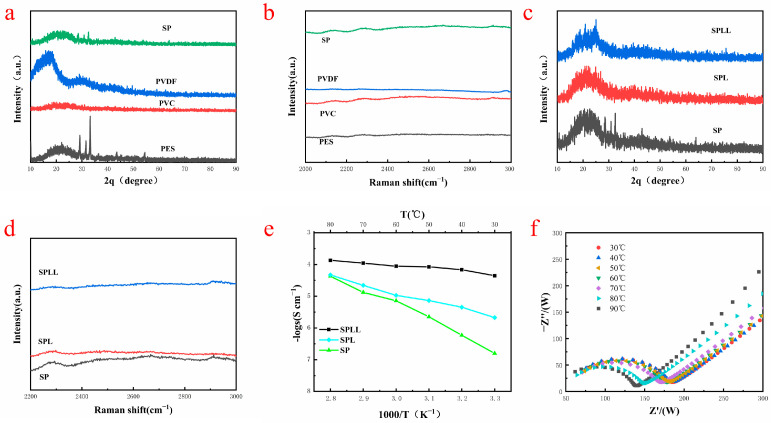
(**a**,**c**) XRD patterns of different polymer. (**b**,**d**) Raman patterns of different polymer. (**e**) Conductivity of polymer electrolyte at different temperatures. (**f**) SPLL impedance diagram at different temperatures.

**Figure 3 nanomaterials-12-03069-f003:**
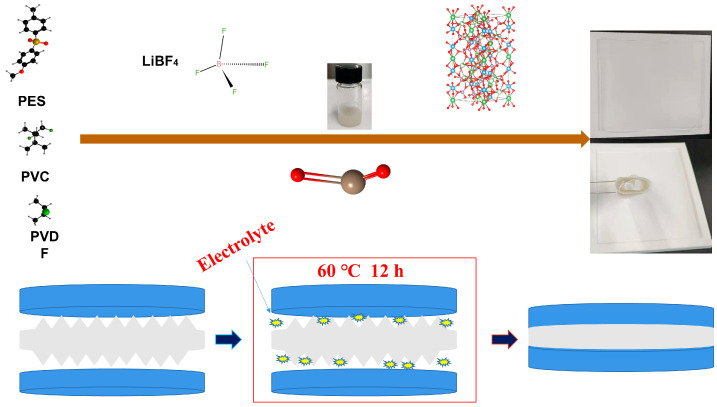
Schematic diagram of preparation of PES-PVC-PVD-LiBF_4_-LATP polymer electrolyte and assembly of solid-state LiNi_0.5_Mn_1.5_O_4_/SPLL/Li lithium battery.

**Figure 4 nanomaterials-12-03069-f004:**
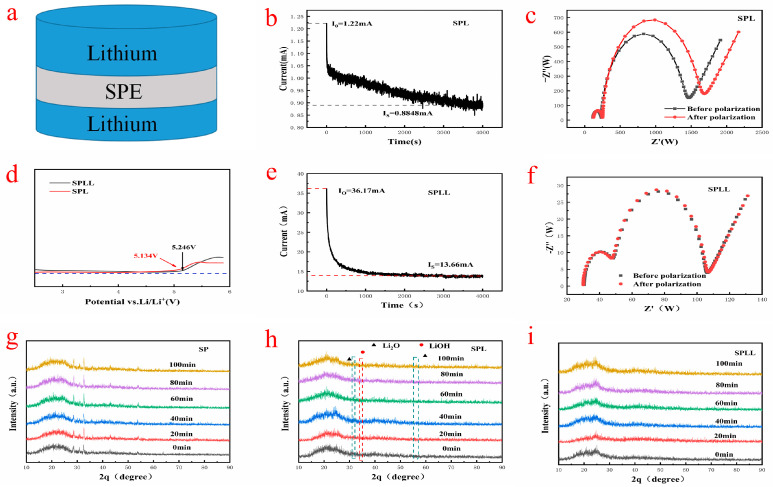
(**a**) Polymer electrolyte membrane assembly test. (**b**,**c**) SPL timing current and impedance before and after timing. (**d**) SPL and SPLL linear voltammetry test. (**e**,**f**) SPLL timing current and impedance before and after timing. (**g**–**i**) XRD patterns after exposing to air (40% relative humidity).

**Figure 5 nanomaterials-12-03069-f005:**
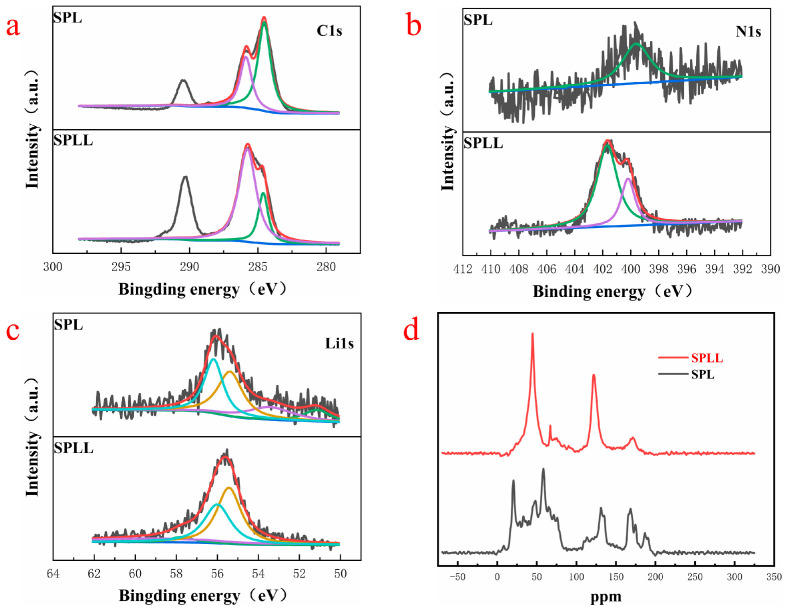
(**a**) Li1s, (**b**) N1s, and (**c**) Li1s XPS spectra of the SPL and SPLL. (**d**) Solid state NMR of the SPL and SPLL.

**Figure 6 nanomaterials-12-03069-f006:**
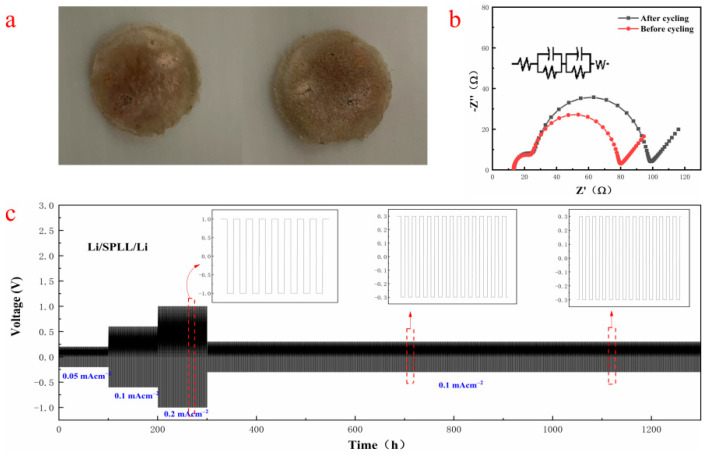
(**a**) Two-sided photo after GCD cycled 1000 times. (**b**) impedance spectra before and after GCD cycle. (**c**) GCD cycling of the Li/SPLL/Li cells at 0.05, 0.1 and 0.2 mA cm^−2^.

**Figure 7 nanomaterials-12-03069-f007:**
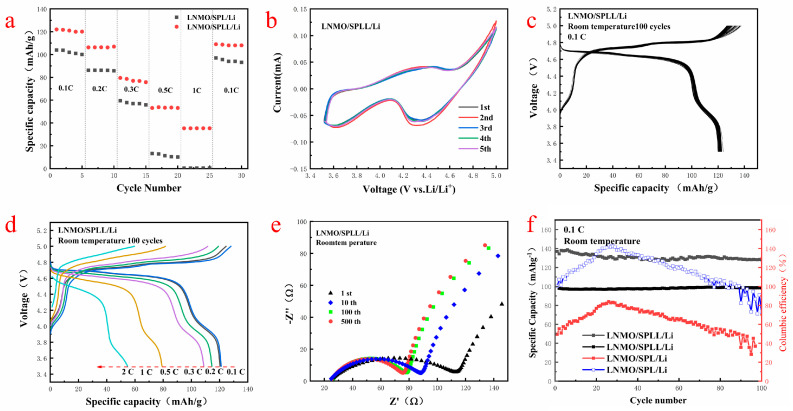
Electrochemical performance of the all-solid-state cell with LNMO cathode and SPLL: (**a**) rate capability; (**b**) CV curves of LNMO/SPLL/Li; (**c**) typical discharge−charge profiles at 0.1 C rate; (**d**) charge/discharge curves under various current densities ranging from 0.1 to 2.0 C; (**e**) Nyquist plots; and (**f**) the cycling performance between 3.5 and 5 V at 0.1 C rate.

**Figure 8 nanomaterials-12-03069-f008:**
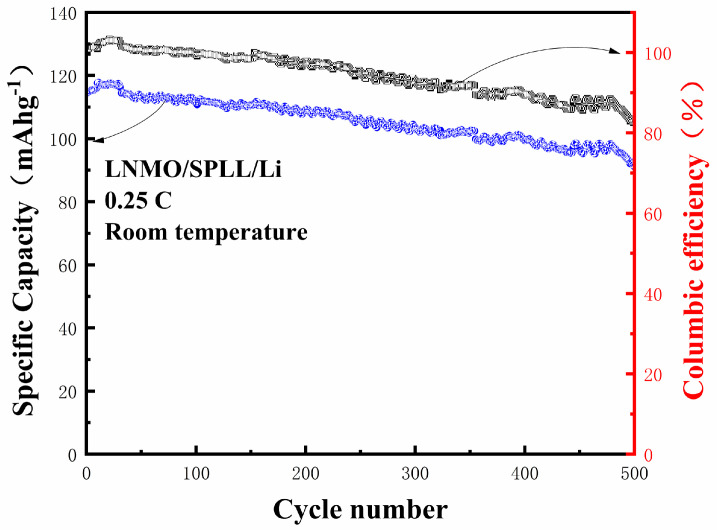
Cycling performance of the assembled all-solid-state LNMO/SPLL/Li battery at 0.25 C.

## Data Availability

Not applicable.

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
