# Peer review of "Preparation and Study of a Simple Three-Matrix Solid Electrolyte Membrane in Air"

_nanomaterials, 2022, doi:10.3390/nano12173069_

Round 1

Reviewer 1 Report

This manuscript reports a composite electrolyte of polymer and inorganic filler for batteries. The results are solid and could be interesting to readers in the field of solid polymer electrolytes or energy storage. I made some suggestions that can improve the paper.

(1) In 2.1: What is the ratio of components, in making LATP powder.

(2) Please specify the abbreviations like SP, SPIL, SPL, SPLL, and SSE.

(3) Please describe in detail how to calculate the capacity of the fabricated LNMO cathode.

(4) (Figure 1 e,h) Provide how you measure the band gap and analysis the graph.

(5) (Figure 1 f,I) Why do the smaller size and smooth surface of particles promote the migration of Li+

(6) (Figure 2e) Please state why SPLL shows less change of conductivity over temperature as compared to others.

(7) (Figure S5) In the SEM image, how do you assert that structure change after the cycle is attributed to a new ion conduction channel, not a negative side effect.

(8) In figure S6: I am wondering why the black color change of electrolyte after cycling would be evidence of close contact between SPLL and LNMO and how it improves electrochemical properties. 

Author Response

Response to Reviewers Comments

  • In 2.1: What is the ratio of components, in making LATP powder.

Response: Thanks to the reviewers for their comments, the improvement of LATP is based on the change of the chemical equation and the doping of the original Al position. The manuscript has been modified for understanding and preparation of materials (page 2).

  • Please specify the abbreviations like SP, SPIL, SPL, SPLL, and SSE.

Response: Thanks to the opinions of the reviewers, SP is an electrolyte membrane formed by stirring and drying three polymers in DMF (among them, DMF 30mL, PVA 1g, PES 2g, PVDF 5% copolymerized base); SPL is added on the basis of SP LiBF4 (PES es: Li=8:1); SPLL is based on SPL with 10%LAZTP added.Modified in the manuscript.

  • Please describe in detail how to calculate the capacity of the fabricated LNMO cathode.

Response: Thanks to the review experts, for the capacity of the cathode material LNMO, we generally follow the theoretical specific capacity of 146.7mAh/g. First, we obtain the mass of the active material in the cathode sheet by weighing and calculation, and then calculate the mass of the manuscript material at different rates according to the formula. The current, using an electrochemical tester, was activated by a small current, followed by charging and discharging tests at different rates.

  • (Figure 1 e,h) Provide how you measure the band gap and analysis the graph.

Response: Thanks to the opinions of the review experts, for the measurement of the band gap, we performed calculation analysis through Materials Studio software to obtain data analysis. For the results of density of state analysis, we obtained the conclusion through literature review and comparative analysis of similar articles.

  • (Figure 1 f,I) Why do the smaller size and smooth surface of particles promote the migration of Li+.

Response: Thanks to the comments of the reviewers, the transport of ions strongly depends on the dense contact of solid particles. These point contacts are very sensitive to the stress generated during electrochemical cycling, which can lead to cracks and poor interfacial contact. Therefore, the smaller and smoother the particle size of LAZTP, the tighter the contact and the smaller the contact stress, which is more conducive to the migration of lithium ions.

  • (Figure 2e) Please state why SPLL shows less change of conductivity over temperature as compared to others.

Response: Because the solid electrolyte LAZTP is added to the SPLL, the polymer electrolyte has a lower glass transition temperature than the solid electrolyte, which is greatly affected by temperature. In addition to the high mechanical strength, the solid electrolyte can also operate in a wider temperature range. maintain a good appearance.

  • (Figure S5) In the SEM image, how do you assert that structure change after the cycle is attributed to a new ion conduction channel, not a negative side effect.

Response: Thanks for the opinions of the review experts, for the experimental conclusions drawn, we tested the batteries by assembling them, and then analyzed the obtained data, and then disassembled the batteries to compare the morphology of the polymer electrolyte membrane, and then reached the conclusion.

  • In figure S6: I am wondering why the black color change of electrolyte after cycling would be evidence of close contact between SPLL and LNMO and how it improves electrochemical properties.

Response: Thanks to the opinions of the reviewers, the black material on the SPLL is the active material LNMO, which indicates that the SPLL is in close contact with LNMO, and the active material and the electrolyte in the solid-solid composite electrode are fully combined, which can shorten the transmission distance of lithium ions, which is conducive to improving the electrochemical performance.

Reviewer 2 Report

This manuscript deals with the preparation and characterization of a novel solid electrolyte with high ionic conductivity and a wide electrochemical window for solid state batteries. In the introduction the authors justify their interest in this topic and include 22 references that may be useful for the readers to understand the fundamentals of the topic and the steps selected by the authors to attain their objectives. In section 2. the authors describe the materials and methods used to prepare the LATP/LAZTP, the HT sintered LATP/LAZTP sheets, and a composite SPLL three-matrix solid electrolyte. Then, the assembly of the solid state LFP/PPLS/Li button cell for testing in the range 3.5-5.0 V, and physicochemical/ electrochemical methods used for the characterization of materials and cells, are also described. In section 3., the experimental results are reported and discussed. XRD confirms the Nasicon structure of LATP/LAZTP. SEM-EDS confirm the incorporation  of Zn into the LAZTP  sample, which causes vacancies in its lattice. EIS shows that the global conductivity of LAZTP is higher than that of LATP, and this is further confirmed by energy bandgap  calculations and density of states , as well as by SEM images.  The interaction mechanism  between lithium ions  and electrolytes is further  revealed by XRD and Raman. EIS at 30-90 C indicates that the polymer electrolyte, SPLL is more adapted  and  more stable  to a broad temperature environment than the SPL. The fabrication process of the battery assembly to be tested under air conditionsis well described and clearly illustrated. CA, AC impedance, LSV are used to calculate the lithium ion migration number and the electrochemical window. It is concluded  that the SPLL can match the high voltage cathode of high-energy-density lithium ion batteries. Examinations by XPS and NMR  of the chemical composition of the polymer electrolyte indicated  that the addition of LAZTP and silica reduces the crystallinity  of SPLL polymer and  enhances  the conductivity of the  material, which is consistent  with the EIS results. GCD cycling of Li/SPLL/Li  cells displays excellent stability even after 1000 cycles of long.term plating /stripping  at constant current density. Further the experimental results combined with  the supplementary information led to more concise  and  precise descriptions  of the studied systems, showing that the novel composite solid electrolyte has high mechanical stability, high ionic conductivity,  serving as a basis for new all-solid-state batteries in  the greenhouse. In summary, a piece of work deserving publication.

Author Response

        Thank you for  for your comments concerning our manuscript. Those comments are all valuable and very helpful for revising and improving our paper, as well as the important guiding significance to our researches. We have studied comments carefully and have made correction which we hope meet with approval. 
